# Uniportal VATS for Diagnosis and Staging in Non-Small Cell Lung Cancer (NSCLC)

**DOI:** 10.3390/diagnostics13050826

**Published:** 2023-02-21

**Authors:** Jone Miren Del Campo, Sergio Maroto, Leyre Sebastian, Xavier Vaillo, Sergio Bolufer, Francisco Lirio, Julio Sesma, Carlos Galvez

**Affiliations:** Department of Thoracic Surgery, Hospital General Dr. Balmis, 03010 Alicante, Spain

**Keywords:** uniportal video-assisted thoracic surgery (VATS), lung cancer, lymph node, staging, mediastinum

## Abstract

Uniportal VATS has become an accepted approach in minimally invasive thoracic surgery since its first report for lobectomy in 2011. Since the initial restrictions in indications, it has been used in almost all procedures, from conventional lobectomies to sublobar resections, bronchial and vascular sleeve procedures and even tracheal and carinal resections. In addition to its use for treatment, it provides an excellent approach for suspicious solitary undiagnosed nodules after bronchoscopic or transthoracic image-guided biopsy. Uniportal VATS is also used as a surgical staging method in NSCLC due to its low invasiveness in terms of chest tube duration, hospital stay and postoperative pain. In this article, we review the evidence of uniportal VATS accuracy for NSCLC diagnosis and staging and provide technical details and recommendations for its safe performance for that purpose.

## 1. Introduction

Non-small cell lung cancer (NSCLC) is the most common type of lung cancer, which is the primary cancer-related cause of death worldwide. Despite the fact that only 30% of lung cancers are diagnosed in their early stages, recent advances in diagnostic techniques (high-resolution CT) and the progressive implementation of screening programmes have increased this proportion substantially with a potential effect on survival [1,2,3].

When dealing with lung cancer suspicion, a pathological diagnosis should be reached first. Traditional methods for diagnosis are flexible bronchoscopy in central nodules or image-guided transthoracic biopsy in peripheral lesions. When a preoperative diagnosis is not reached but there is a high suspicion of malignancy, a diagnostic surgery must be performed. For peripheral nodules, a non-anatomical lung resection (wedge) can provide a diagnosis, and for central lesions, anatomical resections (segmentectomy, lobectomy) might be performed.

Minimally invasive approaches for pulmonary resections have proven safe and feasible and present postoperative advantages [4,5,6], so they are considered the standard nowadays. The development of minimally invasive thoracic surgery has reduced the trauma associated with the surgical approach. In the past few decades, video-assisted thoracic surgery (VATS) has spread worldwide due to technological and educational improvements [7], evolving from the traditional open thoracotomy to minimally invasive surgery with one, two or more incisions. Moreover, robotic-assisted thoracic surgery (RATS) development has progressively gained acceptance within the thoracic surgery community in the last decade [8,9].

In 2004, Rocco et al. [10] described a uniportal VATS access for non-anatomical wedge resection of pulmonary nodules, and in 2011, González-Rivas et al. successfully reported the first uniportal VATS lobectomy for the treatment of early-stage lung cancer [11]. This approach has been employed in most intrathoracic diseases and procedures in which NSCLC is involved, conforming to the oncological surgical standards of the IASLC [12]. Several studies have compared uniportal VATS with other approaches and have shown that there are not significant differences between them in terms of safety [13,14,15]. However, some studies have shown differences in favour of uniportal VATS in terms of postoperative pain, hospital stay and chest tube duration [16,17]. Furthermore, overall and disease-free survival at medium and long follow-up for patients with early-stage NSCLC are not inferior compared to other approaches [18,19,20].

One of the most important prognostic factors in non-small cell lung cancer (NSCLC) is lymph node involvement [21]. An accurate assessment of lymph node involvement is needed to determine which patients will benefit from straightforward surgical treatment and which will benefit from a multimodal approach or non-surgical treatment within multidisciplinary tumour boards.

Imaging tests are the first-line staging tests to be performed in lung cancer. Suspicious lymph nodes detected on CT or PET must be confirmed with invasive assessment. For central lesions, tumours > 3 cm in diameter or less than 1.5 cm with low uptake in PET (lepidic), tumours invading surrounding structures, or N1 positive in PET, an invasive pathological staging must be performed as well [22,23].

There are different techniques to pathologically assess mediastinal lymph nodes. The first one is the transbronchial biopsy, with the more recent use of ultrasound-guided transbronchial or transoesophageal biopsy (EBUS/EUS), and if there is no confirmation but high suspicion [23,24,25], the next step is the surgical biopsy that has been traditionally performed through mediastinoscopy and anterior mediastinotomy. With the development of minimally invasive surgery, diagnostic lymphadenectomy through video-assisted surgery (VATS) has become a safe and reliable test to perform as a second-line method for mediastinal lymph node assessment [26,27]. 

The aim of this paper is to review uniportal VATS as a diagnostic method for solitary suspicious nodules and for assessing lymph node involvement in patients with NSCLC.

## 2. Methods

Electronic searches were performed using the PubMed database. Keyword and MeSH term searches were performed for 4 groups: (1) “uniportal*” or “single port*” or “single incision*”; (2) “VATS” or “thoracosp*” or “video assist*”; (3) “staging*” or “diagnose*”; (4) “lung cancer*” or “NSCLC*”. The 4 groups were combined using the boolean operator “AND”.

Abstracts and titles were used for the initial screening of all search results. Selected studies were defined as those that used a uniportal VATS approach to diagnose suspicious nodules or to stage NSCLC. All studies were restricted to humans only. Studies were excluded if the primary focus was on anaesthetic techniques (e.g., non-intubated or awake surgery). 

After the initial screening, the full text of the rest of the potentially relevant articles was retrieved. Additional potentially relevant studies were identified by examining the reference lists of all retrieved articles and the online list of articles citing them. The full texts of all these studies were then critically appraised and selected based on their quality and relevance to the primary objective of defining the role of uniportal VATS in diagnosing and staging lung cancer.

## 3. Diagnosis of NSCLC

High-resolution CT and lung-cancer screening programmes have increased the number of pulmonary nodules that require further evaluation. Solid nodules > 8 mm in diameter and part-solid nodules with solid component ≥ 6 mm require more accurate diagnostic techniques such as PET-CT and biopsy [28,29,30].

An initial pathological diagnosis is mandatory whenever lung cancer is suspected to determine the optimal treatment. Adequate tissue samples must be obtained for a definitive diagnosis and for testing biomarkers with predictive and prognostic significance.

Initially, a fiberbronchoscopy is recommended, especially for diagnosing central nodules, but it can also be useful for diagnosing peripheral pulmonary lesions. The global diagnostic accuracy of conventional bronchoscopy for lung cancer diagnosis is low, but the sensitivity for central lesions can be up to 88% due to their nearness to the main and lobar bronchi. However, the diagnostic accuracy for peripheral lesions with a diameter of 2 cm or less shows a low sensitivity of only 34%. The complication rate of diagnostic fiberbronchoscopy is low, but it can include pneumothorax (1%) and endobronchial haemorrhage (from 0.6 to 5.4%) in some series [31,32]. Electromagnetic bronchoscopic navigation is also available, which has an estimated diagnostic yield of 65% to 73% [33]. As a technological evolution of conventional fiberbronchoscopy, radial EBUS-guided transbronchial biopsy shows diagnostic rates that are non-inferior to those of computed tomography (CT)-guided biopsies but presents lower complication rates. Its overall accuracy ranges between 63 and 80%, irrespective of the size and location of the lesion [34]. 

Image-guided transthoracic procedures are frequently used for managing suspicious pulmonary nodules, particularly in cases of peripheral lesions where flexible bronchoscopy performances are lower. The overall sensitivity of transthoracic needle aspiration (TTNA) for the diagnosis of peripheral lung cancer is 90% [35]; therefore, it is one of the main methods used for diagnosis nowadays.

In cases where conventional techniques have not reached a diagnosis but the clinical and radiological suspicion is high, surgical resection with diagnostic intent is the last option. If the lesion is amenable to non-anatomical surgical wedge resection, minimally invasive approaches, including uniportal VATS, can be considered [36]. For more central lesions, an anatomical segmentectomy or lobectomy through a minimally invasive approach should be performed. Hybrid operating rooms are also available in some hospitals where cone-beam CT-guided surgery can be performed to increase the detection of nodules that are not easily palpable or visible [37].

### Uniportal VATS for NSCLC Diagnosis

This approach has been shown to reduce rates of postoperative pain, hospital stay and chest tube duration [15]. 

The patient should be placed in a stable lateral decubitus position to avoid injuries to the brachial plexus or shoulder. We have used two techniques to slightly enlarge the intercostal space: operating tables that can be flexed and the use of an inflatable pneumatic roller/semirigid roller behind the contralateral side. This mild contralateral flexion makes it easier to manipulate instruments through the incision during the procedure.

An incision of 3–4 cm in length in the fifth intercostal space in the anterior axillar line is performed. To make the procedure easier, a wound retractor can be placed. A 5 mm or 10 mm 30° video thoracoscope is used in combination with an endo-grasper and an endo-stapler. Peripheral lesions can be easily identified, especially if visceral pleura retraction is present. For non-visible lesions, after careful review of the CT images within the 3D planes, an attempt should be made to palpate the nodule through the incision. The inability to identify the lesion targeted for resection increases to 63% if the nodule is smaller than 10 mm or is located more than 5 mm from the pleural surface. Several image-guided preoperative localization techniques have been developed to aid in intraoperative nodule identification during VATS, such as dye marking, hook wire or microcoil placement [38] (Figure 1), and new technological advances, namely electromagnetic navigation bronchoscopy (ENB) [39] and virtual-assisted lung mapping (VAL-MAP) [40] offer excellent outcomes. 

Lesion margins can be marked with monopolar cautery on the lung surface, and then a wedge resection can be performed, if feasible, by using parenchymal endo-staplers to ensure adequate safety margins. In the uniportal VATS approach, the placement of the instruments within the wound is important: the thoracoscope should be placed on the superior part of the incision; in the central portion, a grasper retracts the lung for exposure; and in the most inferior part, the endo-stapler is placed [10,27,36,41] (Figure 2) (Appendix A). In order to avoid collision between the instruments, the lung-retracting grasper and endo-stapler can be crossed in an X-shape for better space utilization. Figure 3 shows a step-by-step uniportal VATS diagnostic wedge resection (Figure 3).

Diagnostic surgical resection allows for the specimen to be obtained, including the nodule and safety margins, and enables further pathological, immunohistochemical and biomolecular testing. After intraoperative frozen-section study, lobar or sublobar anatomical resection must be completed whenever NSCLC is diagnosed in the clinical early stage, in combination with lobe-specific or systematic lymph node dissection according to the IASLC criteria [43]. Wedge resection combined with lymphadenectomy remains a controversial option regarding the available evidence but can be considered mainly for ground-glass opacities (GGOs) or part-solid lesions with a consolidation-to-tumor ratio of less than 0.5 and a size of less than 2 cm in diameter [44,45]. In highly suspicious nodules not amenable to wedge resection, initial segmentectomy can be considered as an alternative with further frozen-section analysis [46]. Resection of lung parenchyma for suspicious opacities without preoperative tissue diagnosis is rare (2.1%) and has a satisfactory safety rate with very low postoperative morbidity and mortality. The diagnostic accuracy of frozen-section analysis for tumours with a diameter of less than 1 cm and greater than 1 cm has been shown to be 79.6% and 90.8%, respectively. If it is inconclusive, the decision must be made by the surgeon in charge, who may wait for the definitive anatomy and, if the resection was inadequate, reopen the patient or perform an anatomical resection on the assumption that it may be a benign lesion [47,48].

## 4. Uniportal VATS for NSCLC Staging

The staging of lymph node involvement is recognised as a key aspect of the initial management of NSCLC that allows for adequate treatment selection and predicts outcomes. 

Multiple studies have examined the lymphatic drainage pathways of the lung and have analysed the influence of N1 and N2 lymph node involvement in 5-year overall survival (OS) with a progressive worse prognosis regarding mediastinal (N2) and multiple stations involvement (Table 1) [49,50,51,52].

In fact, lymph node (LN) status is one of the strongest prognostic parameters for patients with NSCLC. Imaging tests such as CT and PET scans combined with surgical assessment have become standard components of the initial clinical staging.

Contrast-enhanced thoracic and abdominal CT scans have high specificity (over 80%) but only moderate sensitivity (60%), so a diagnosis of metastatic spread cannot rely exclusively on them [53]. Positron emission tomography (PET) should also be performed, but its sensitivity varies according to the literature, ranging from 75 to 91%, with its specificity ranging from 78 to 93%. Studies have shown that PET is less specific in cases of enlarged lymph nodes and, on the other hand, less sensitive in normal-sized lymph nodes less than 1 cm in diameter. In clinical stage I NSCLC, positive mediastinal lymph nodes require biopsy confirmation since the positive predictive value of PET is unacceptably low in these cases. By contrast, a negative PET for mediastinal assessment in clinical stage I allows for surgery with no further examinations [22,23].

Therefore, invasive assessment of lymph node status is recommended in all patients except those with small (≤3 cm) peripheral carcinomas without evidence of nodal enlargement on computed tomography (CT) or increased uptake on positron emission tomography (PET). The first one is the transbronchial/transcarinal biopsy, which in specialised centres can achieve a sensitivity of 76% and a specificity of 98%, which means that, after a negative result in highly suspicious cases, a surgical biopsy should be performed. With the additional use of endobronchial-ultrasound needle aspiration (EBUS-NA) or endoscopic ultrasound needle aspiration (EUS-NA), sensitivity is increased up to 80–90%. However, it decreases if lymph nodes are smaller than 5 mm, if the test is not gently performed or if the staff is not technically experienced [23,24,25].

Surgical biopsy has been traditionally performed through mediastinoscopy or anterior mediastinotomy. Mediastinoscopy allows reaching most of the mediastinal lymph node stations bilaterally, especially those located paratracheally, subcarinally and surrounding the main bronchi. However, there are some stations, such as the aortopulmonary window, para-aortic, para-oesophageal or pulmonary ligament stations, that are beyond the scope of this test unless more complex technical variations such as video-assisted mediastinal lymphadenectomy (VAMLA) or transsternal extended mediastinal lymphadenectomy (TEMLA) are performed. Anterior mediastinotomy is a technique used to biopsy stations far from peritracheal areas such as the left para-aortic and the aortopulmonary lymph nodes [54], but its use has decreased significantly since the advent of VATS. 

With the development of minimally invasive surgery, safe and reliable diagnostic VATS lymphadenectomy of all ipsilateral stations can be performed whenever there is suspicion of unproven nodal involvement or to rule out occult lymph node metastasis, depending on the clinical TNM stage [26,55]. VATS shows a sensitivity for lymph node assessment ranging from 58 to 100% (median 99%) and a false negative rate of 4% [56]. Uniportal VATS is feasible and has been demonstrated to be at least as safe as multiportal VATS for lymph node assessment [57].

### 4.1. Position and Lymph Node Dissection

Technically, the patient should be placed as explained before. For either the right or left procedure, a 3–4 cm incision is made in the fifth intercostal space at the anterior axillary line. The assistant places the 30° thoracoscope at the top of the incision. There are two accepted types of mediastinal lymph node dissection: Sampling: it involves the removal of one or more lymph nodes based on preoperative findings.Systematic nodal dissection (MLND): it involves the removal of all mediastinal tissue, including the lymph nodes located within anatomical landmarks.

The longer overall survival after systematic lymphadenectomy is still controversial. A recent meta-analysis of five randomised controlled trials suggested that mediastinal nodal dissection might provide better survival and recommended further evaluation of this question through multicentre trials [58,59,60,61].

### 4.2. Instrumentation

Useful instruments for MLND include a curved-tip metallic endoscopic suction, a double-jointed node grasper and an energy device (monopolar, advanced bipolar, ultrasonic). It is important to keep the field as clear as possible from blood by using blunt dissection with the suction and performing dissection and hemostasia with the energy device. There are two different techniques: The “grasping” technique is performed using a grasper to grasp the target lymph nodes.The “non-grasping” technique essentially consists in performing an MLND by using only energy devices and a metallic endoscopic suction, avoiding directly grasping the lymph nodes in order to keep the capsule intact.

The standard method of MLN resection has not yet been established. There is a controversy between these two techniques. Compared with the traditional “grasping” MLND, the non-grasping strategy can avoid damage to LNs and ensure the integrity of LNs, thus preventing possible tumour spread. Moreover, *en bloc* dissection of LNs and fatty tissue avoids leaving unnoticed LNs in the studied station [62].

### 4.3. Surgeon’s Positioning

The patient should be positioned as explained in the diagnostic section. The surgeon for most stations is positioned as explained above, but there are some stations where changing position with the assistant could be helpful. For the pulmonary ligament, perioesophageal and subcarinal stations, the assistant’s preferred position is more cranial, very close to the arms of the patient, so that the surgeon can operate on the lower part of the chest without constantly obstructing the thoracoscope. Other surgeons prefer the assistant to stand on the opposite side of the patient, but it takes longer to develop the skills to assist the camera in a mirror image. The use of 30-degree thoracoscopes makes it easier to avoid constant collisions between the surgeon’s instruments and the camera. The scrub nurse stands on the opposite side of the patient, close to the legs, so she/he can visualise a second screen behind the surgeon and easily assist both the surgeon and the assistant [42,63,64,65,66,67,68] (Figure 4).

### 4.4. Nodal Stations

In order to perform an accurate mediastinal lymph node dissection, a comprehensive knowledge of mediastinal nodal stations and their corresponding anatomical landmarks is essential [43,69,70] (Table 2).

### 4.5. Technical Description

Right side (**Appendix A**):(a)Upper- and lower-right paratracheal stations (2R and 4R) (Figure 5): Procedure: First, divide the mediastinal pleura at the intersection of the azygous vein with the superior vena cava. Retracting the pleura with the suction and energy device, continue dividing the pleura over the azygous vein between the vagus nerve in the left and the superior vena cava. Next, using suction or a grasper, start removing all the fat and lymph nodes *en bloc* along the anterolateral side of the trachea, just below the caudal edge of the right brachiocephalic artery. Sometimes, the phrenic nerve runs more lateral to the superior vena cava and not over it; before deepening the dissection towards the posterior aspect of the superior vena cava, it is necessary to check the position of the nerve to avoid accidental damage. The most common complication of a complete paratracheal dissection is chylothorax, or bleeding from small vessels.Recommendations: Part of station 4R lies below the azygous vein, in continuity with station 10R. To facilitate dissection, once the lymphadenectomy at station 10R has been performed, the azygous vein can be dissected with the aid of a dissector, and a vessel-loop can be used to pull it to widen the view and facilitate access to this area.(b)Prevascular (3A): Procedure: It is an uncommon station that includes lymph nodes that are situated anterior to the phrenic nerve in the pericardial fat. Remove the pericardial fat and lymph nodes without damaging the phrenic nerve.Recommendations: Tilt the operating table posteriorly away from the surgeon so that the lung falls slightly backwards. Making use of table rotation can make it easier for the lung to remain away from the field without the need for traction.(c)Subcarinal (7R): (Figure 6)Procedure: After dividing the posterior mediastinal pleura from the pulmonary ligament to the right main bronchus, the operating table is turned anteriorly towards the surgeon so that the lung falls slightly anteriorly. The surgeon then pulls the lung towards him/her with a sponge stick. This station must be explored in depth in order to remove all the subcarinal space and not just the lymph nodes below the intermediate bronchus. The criteria from the IASLC for complete resection (R0) standards always include dissection of the subcarinal station. During this dissection, the contralateral left main bronchus can be visualised. The right bronchial artery runs from the descending aorta to the right main bronchus, crossing the subcarinal space.Recommendations: Sometimes it is useful to have the assistant pull up the pleura or oesophagus with the suction to have a better view and a more comfortable situation. Depending on the patient’s anatomy, it may be useful to position the assistant on either side of the surgeon.(d)Paraoesophageal and pulmonary ligament stations (8–9R): (Figure 7)Procedure: In order to access the lower stations, the lower lobe is retracted cranially and anteriorly to allow exposure of the inferior pulmonary ligament. The thoracoscope and the retractor can be held by the assistant while the surgeon uses bimanual instrumentation to divide the inferior pulmonary ligament. Pulmonary ligament lymph nodes can be located here, in close contact with the inferior pulmonary vein. Paraoesophageal nodes usually lie next to the oesophageal side or in the pericardial-oesophageal groove. As in the subcarinal space, pulling up the oesophagus with the suction (by the assistant) provides optimal visualisation and a more comfortable situation. Special care must be taken with the thoracic duct, which lies between the oesophagus and the aorta at this location.Recommendations: The assistant is positioned more cranially, very close to the patient’s arms, so that the surgeon can dissect the lower part of the chest without constantly crashing the thoracoscope. Other consultants prefer the assistant to be positioned on the opposite side of the patient, but it takes longer to develop the skills to assist the camera in a mirror image. In small thoracic cavities or with a voluminous diaphragm, it is useful to push it away caudally with the help of the assistant.Left side (**Appendix A**):(a)Prevascular (3AL): (Figure 8) Procedure: Similar to the right side, remove the pericardial fat and lymph nodes located anterior to the phrenic nerve. This station can be particularly difficult during uniportal VATS due to its anterior location.Recommendations: As described earlier, using table rotation can make it easier to keep the lung away from the field without the need for traction.(b)Lower paratracheal (4L): (Figure 9)Procedure: The left paratracheal station is placed deep inside the aortopulmonary window and is a surgical challenge for systematic lymph node dissection. The recurrent laryngeal nerve can be damaged, and, in some cases, the ligamentum arteriosum can hinder exposure. Station 4L can be easily reached and excised up to the point of visualising the left side of the trachea and left main bronchus. The dissection starts at the main bronchus under the aorta and continues until the left tracheobronchial angle is reached. Some authors recommend avoiding energy devices to prevent neural damage [71,72].Recommendations: Lymphadenectomy at stations 5 and 6 can facilitate access to this station. The assistant can pull up the aorta with the suction in order to provide better visualisation of this area.(c)Aortopulmonary window and para-aortic (5–6): (Figure 10) Procedure: It is possible to dissect this area with the assistant standing to the right of the surgeon. After initial division of the mediastinal pleura above the upper lobe vein, dissect the fat tissue and lymph nodes that are below the aortic arch and posterior to the phrenic nerve. By using a sponge stick, gently retract the upper lobe caudally. The assistant can hold both the suction and the grasper, allowing the surgeon the possibility of bimanual dissection. Recommendations: Care should be taken to avoid damage to the recurrent laryngeal nerve when dissecting lymph nodes below the aortic arch. Consider the balance of benefits and risks from systematic dissection and the chance of potential laryngeal nerve damage.(d)Subcarinal (7L): (Figure 11)Procedure: Perform as described in 7R. Note that the vagus lies behind, the aorta is above and the oesophagus runs deeper than on the right side.Recommendations: Sometimes it is useful to have the assistant pull up the aorta on the left side with the suction to have a better view and a more comfortable situation. Depending on the patient’s anatomy, it may be useful to position the assistant on either side of the surgeon. (e)Paraoesophageal and pulmonary ligament (8–9L): (Figure 12)Procedure: Performed as described in 8–9R. These stations can be safely dissected while dividing the pulmonary ligament and are easier to dissect with the assistant standing to the left of the surgeon, cranially and close to the patient’s arms.Recommendations: In small thoracic cavities or with a highly inserted diaphragm, it is useful to push it away with a suction or sponge stick to clearly visualise the paraoesophageal groove.

## 5. Conclusions

When conventional diagnostic techniques fail and surgery is required, uniportal VATS is a feasible and safe minimally invasive alternative with the advantage of providing the surgeon with an open, approach-like view. 

If a pulmonary nodule is highly suspicious of being a malignant tumour, the uniportal VATS approach allows for intraoperative diagnosis through wedge resection and, if confirmed, for the completion of anatomical resection and accurate lymph node dissection.

Accurate lymph node dissection is essential for pathological staging, prognosis assessment and multimodal treatment indications in locally advanced NSCLC. 

With regard to mediastinal staging, the advantages of uniportal VATS over traditional mediastinoscopy are additional access to limited stations (5, 6, 8, 9) and the possibility of subsequent lung resection.

## Figures and Tables

**Figure 1 diagnostics-13-00826-f001:**
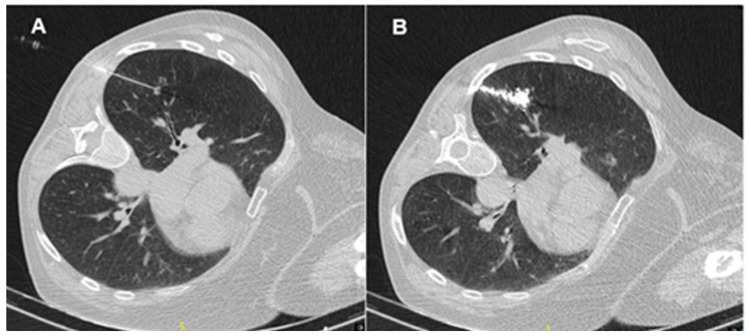
(**A**): CT-guided preoperative localization technique with needle; (**B**): CT-guided preoperative localization technique with microcoil marking in the nodule.

**Figure 2 diagnostics-13-00826-f002:**
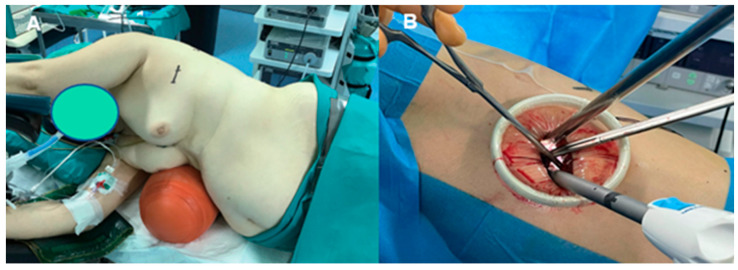
(**A**): Patient positioning during left VATS with an inflatable roll behind the contralateral side; (**B**): instrumentation: during most of the procedure, the camera is at the top of the incision and uniportal instruments and endo-staplers are at the bottom [42].

**Figure 3 diagnostics-13-00826-f003:**
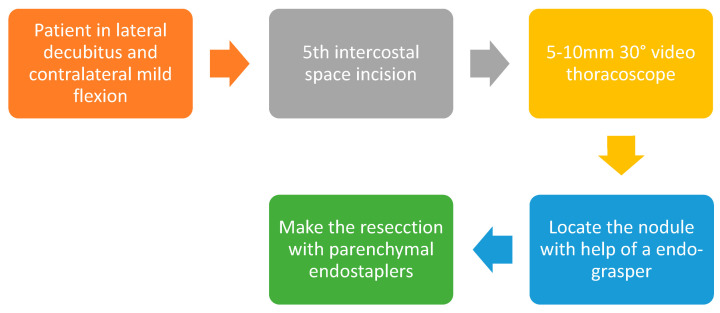
Uniportal VATS diagnostic wedge (step by step).

**Figure 4 diagnostics-13-00826-f004:**
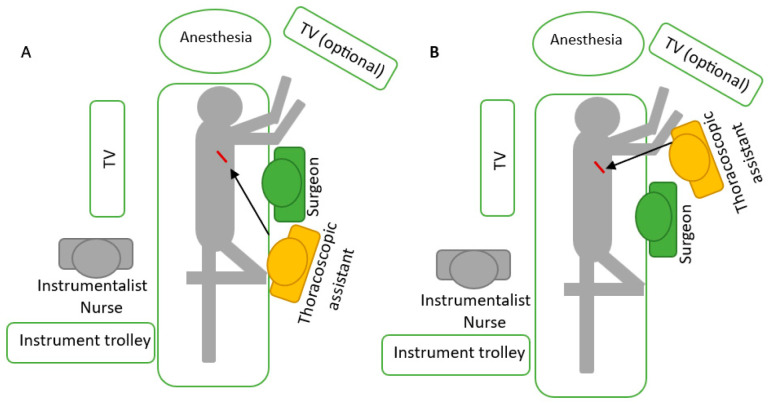
Position of the surgeon, camera assistant and scrub nurse. (**A**) The assistant stands caudally to the surgeon during most steps of the procedure; (**B**) for certain steps in the lower part of the chest, the assistant stands cranially to the surgeon to avoid constant collision with his instruments [64].

**Figure 5 diagnostics-13-00826-f005:**
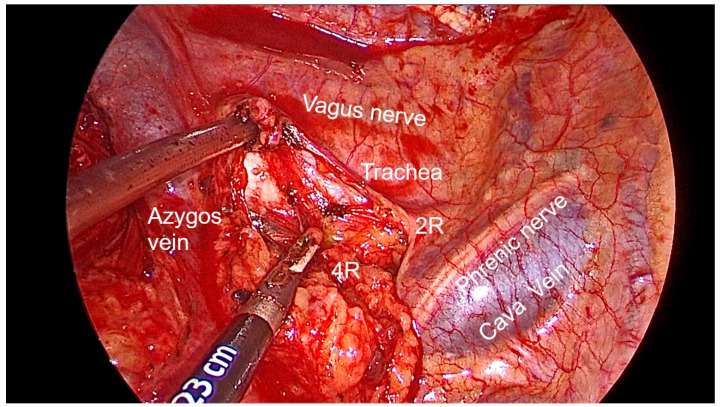
Anatomic landmarks during right paratracheal lymph node dissection (2R and 4R).

**Figure 6 diagnostics-13-00826-f006:**
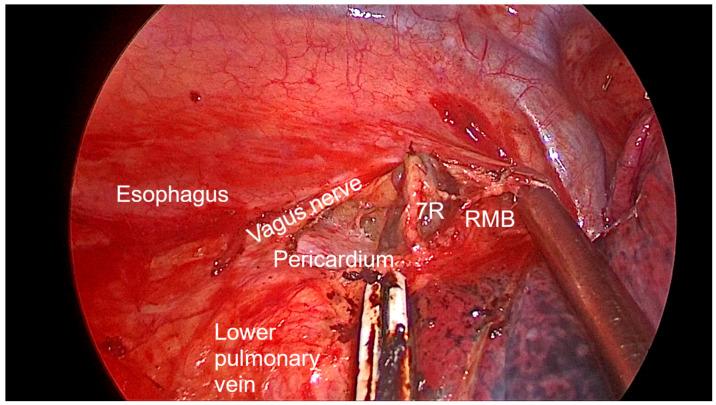
Anatomic landmarks during right subcarinal lymph node dissection (7R). RMB—right main bronchus.

**Figure 7 diagnostics-13-00826-f007:**
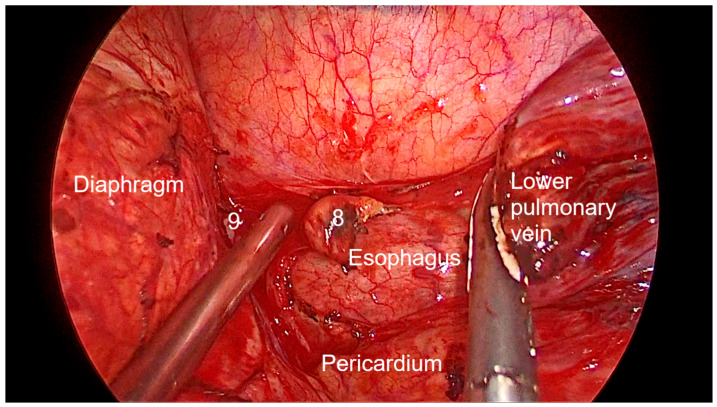
Anatomic landmarks during right paraoesophageal (8) and pulmonary ligament (9) lymph node dissection.

**Figure 8 diagnostics-13-00826-f008:**
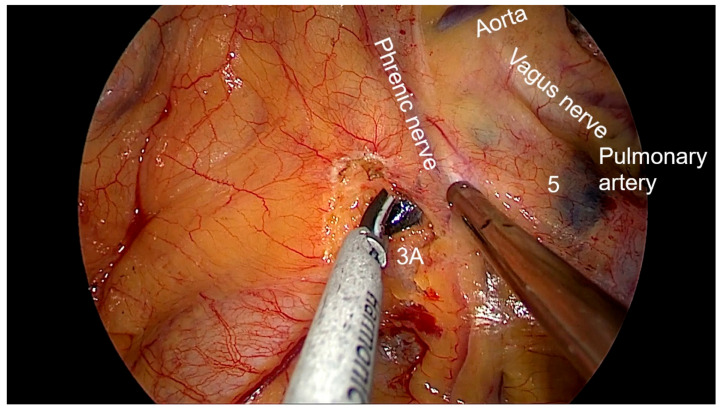
Anatomic landmarks during left prevascular lymph node dissection (3A).

**Figure 9 diagnostics-13-00826-f009:**
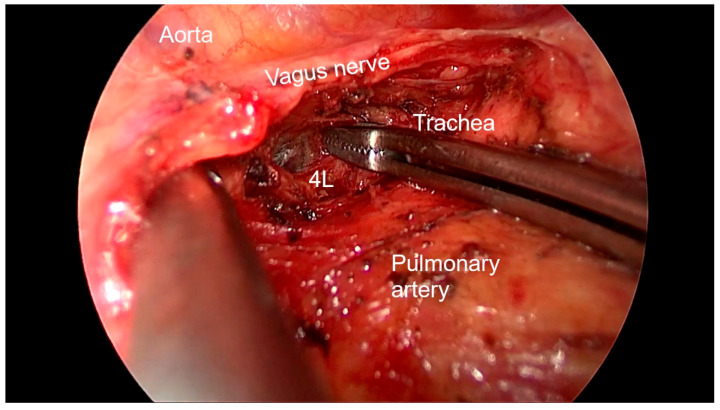
Anatomic landmarks during left paratracheal lymph node dissection (4L).

**Figure 10 diagnostics-13-00826-f010:**
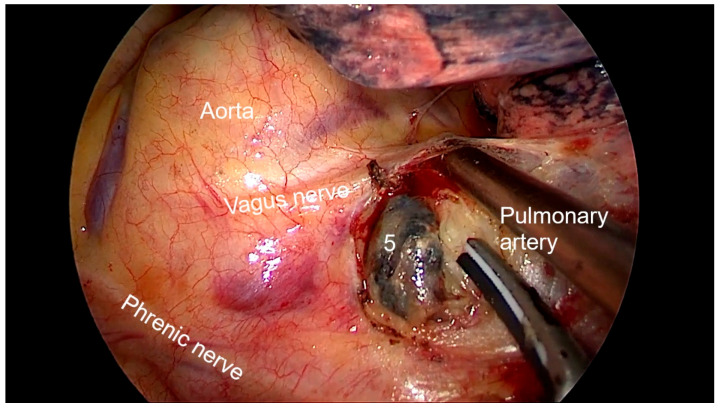
Anatomic landmarks during left aortopulmonary window (5) and paraaortic (6) lymph node dissection.

**Figure 11 diagnostics-13-00826-f011:**
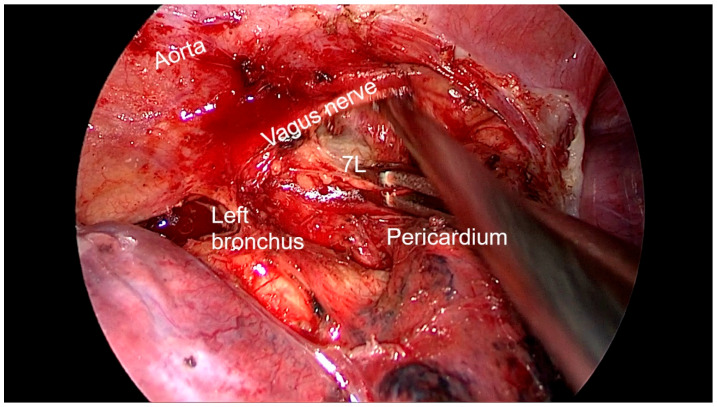
Anatomic landmarks during left subcarinal lymph node dissection (7L).

**Figure 12 diagnostics-13-00826-f012:**
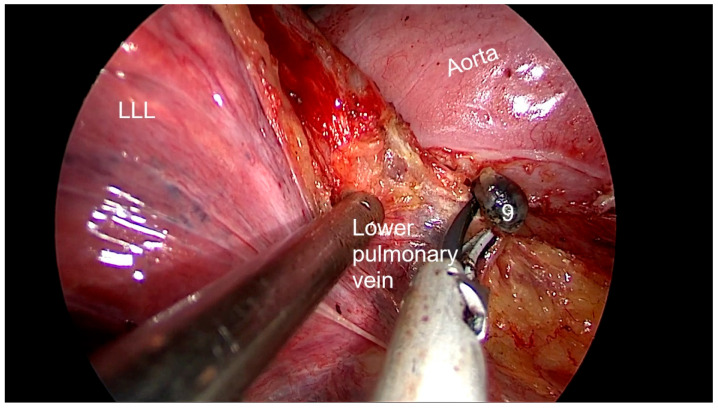
Anatomic landmarks during left pulmonary ligament lymph node dissection (9).

**Table 1 diagnostics-13-00826-t001:** N1 and N2 involvement in OS.

Lymph Node Involvement	5-Year Overall Survival (%)
N0	63.1%
N1a (single N1 zone)	48%
N1b (multiple N1 zones)	35%
N2a (single N2 zone)	34%
N2b (multiple N2 zones)	20%

Influence of N1 and N2 lymph node involvement in 5-year overall survival (OS) [44,45,46,47].

**Table 2 diagnostics-13-00826-t002:** Lymph node station landmarks.

Lymph Node Compartments	Lymph Node Stations	Anatomical Limits
*Superior mediastinum*	2R	Superior: thoracic inlet
		Inferior: intersection of the caudal margin of the left brachiocephalic artery (BCA) with the trachea
		Left: left (2L) and right (2R) are divided along the midline of the trachea
		Right: mediastinal pleura
		Anterior: superior cava vein (SVC)
		Posterior: trachea
	3A	Superior: thoracic inlet
		Inferior: carina
		Left: external posterior wall
		Right: anterior border of the SVC
		Anterior: mediastinal pleura
		Posterior: pericardium
	4R	Superior: intersection of the caudal margin of the left BCA with the trachea
		Inferior: inferior border of the azygos vein
		Left: left (4L) and right (4R) are divided along the midline of the trachea
		Right: mediastinal pleura
		Anterior: SVC
		Posterior: trachea
	4L	Superior: superior border of the aortic arch
		Inferior: carina
		Left: medial to arterial ligament
		Right: left (4L) and right (4R) are divided along the midline of the trachea
		Anterior: trachea anterior wall
		Posterior: trachea posterior wall
*Subaortic compartment*	5	Superior: inferior border of the aortic arch
		Inferior: superior border of the left pulmonary artery
		Left: mediastinal pleura
		Right: plane between the arterial ligamentum and the left vagus nerve
		Anterior: left phrenic nerve
		Posterior: left vagus nerve
	6	Superior: upper border of the aortic arch
		Inferior: lower border of the aortic arch
		Left: mediastinal pleura
		Right: ascending aorta
		Anterior: ascending aorta anterior wall
		Posterior: ascending aorta posterior wall
*Subcarinal and inferior* *compartment*	7	Superior: carina
		Inferior: upper border of the lower lobe bronchus on the left; lower border of the bronchus intermedius on the right
		Left: left main bronchus mediastinal wall
		Right: right main bronchus mediastinal wall
		Anterior: pericardium
		Posterior: oesophagus
	8	Superior: upper border of the lower lobe bronchus on the left; lower border of the bronchus intermedius on the right
		Inferior: diaphragm
	9	Superior: inferior pulmonary vein
		Inferior: diaphragm

Anatomical landmarks of mediastinal lymph stations [68].

## Data Availability

No new data were created or analysed in this study. Data sharing is not applicable to this article.

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
