# Peer review of "Uniportal VATS for Diagnosis and Staging in Non-Small Cell Lung Cancer (NSCLC)"

_diagnostics, 2023, doi:10.3390/diagnostics13050826_

Round 1

Reviewer 1 Report

The aim of the authors is to review uniportal VATS as a diagnostic method for solitary suspicious nodules and for assessing lymph node involvement in patients with NSCLC. The techniques are thoroughly described, and well supported by figures and videos. Considering the spread of these techniques and the effective approach suggested, I’m convinced that it would be worth to let the readers know about it. Nevertheless, I would suggest to also consider awake surgery techniques as a further trick in uniportal VATS.

Author Response

Thank you for your comments, the authors have not wanted to comment on the technique of uniportal vats in the awake patient because we believe that it is outside the scope of the article, since at no time as such do we talk about the anaesthetic technique used. But it could be interesting for other publications in which the anaesthetic technique and its risks and benefits are emphasised.

Reviewer 2 Report

In the present paper Authors report some evidence about the accuracy of uniportal VATS in lung cancer diagnosis and staging. No methodology of the reviewing process is reported. The paper is missing about a few of crucial information such as a comparison between traditional and uniportal VATS. The use of the hybrid operative room also need to be cited when discussing intraoperative localization technique for non-palpable lung nodules (iVATS).

Major revisions:

- Authors should report on the reviewing process

- Authors should discuss potential advantages and/or drawbacks of uniportal VATS over other traditional VATS techniques, reporting evidences from literature

- Authors should cite iVATS as an option for intraoperative localization of undiagnosed non-palpable lung nodules, discussing the potential application of uniportal VATS in a hybrid environment (in my opinion feasible).

Minor revisions:

- page 3, line 100: please do not state that “surgical resection with diagnostic intention” is mandatory, it could be an option.

- page 3, line 103: when reporting on anatomical resection for central undiagnosed lesions please discuss the risk of performing unnecessary major surgery for a potentially benign disease, reporting benign rates from literature.

- page 3, line 131: Authors should state how they consider safety margins adequate.

- page 14, line 402-403: please remove these lines from the conclusion section or better explain them. Authors cannot compare unilateral uniportal VATS and tradition al mediastinoscopy (a bilateral procedure) in terms of mediastinal staging. Only a bilateral uniportal VATS approach (not recommended) could be compared with mediastinoscopy’s staging yields.

Author Response

Major revisions:

- Authors should report on the reviewing process--> Added

- Authors should discuss potential advantages and/or drawbacks of uniportal VATS over other traditional VATS techniques, reporting evidences from literature --> The only advantage of uniportal VATS over conventional VATS is less postoperative pain and shorter length of stay, which we have already mentioned in the article in references 13-14. 

- Authors should cite iVATS as an option for intraoperative localization of undiagnosed non-palpable lung nodules, discussing the potential application of uniportal VATS in a hybrid environment (in my opinion feasible). --> Added

Minor revisions:

- page 3, line 100: please do not state that “surgical resection with diagnostic intention” is mandatory, it could be an option.--> Changed

- page 3, line 103: when reporting on anatomical resection for central undiagnosed lesions please discuss the risk of performing unnecessary major surgery for a potentially benign disease, reporting benign rates from literature.--> Added

- page 3, line 131: Authors should state how they consider safety margins adequate.--> In this case we have not specified more than the margin because, as it is a diagnostic wedge, it is sufficient that the lesion is integrated in the wedge, since if NSCLC is diagnosed, the anatomical resection will be completed. There is an exception for GGO but this is explained below in the article.

- page 14, line 402-403: we have changed to ''In terms of mediastinal staging, the advantages of uniportal VATS over traditional mediastinoscopy are additional access to limited stations (5, 6, 8, 9) and the possibility of subsequent lung resection.''

Reviewer 3 Report

Line 13: "in experience hands": this phrase should better be avoided

Lines 27-33: Electromagnetic navigation is also a modality, please comment. 

Lines 106-128: This introduction is too long to finally state that wedge resection is feasible by uVATS

Line 146: Frozen section: what is the diagnostic yield for pure GGOs ans semi-solid nodules? What if frozen section is inconclusive? 

Lines 218-373: This discussion is redundant, the lymphadenectomy and the anatomical landmarks are the same regardless of the access. In addition, the message is not clear; Do the authors prefer uVATS for mediastinal staging than mediastinoscopy? And after staging, do the return to the thorax at a later time in order to perform the anatomical resection  

Author Response

Line 13: deleted

Lines 106-128: This is not an introduction to how a wedge is performed, but is intended to explain that when you need to perform a diagnostic resection you may find yourself in the situation of not locating the nodule and the tools you can use to make the surgery successful.

Lines 27-33: Add in line 89

Line 146: The diagnostic accuracy of FS for tumors ≤ 1 cm and larger than 1 cm in diameter was 79.6% and 90.8%, respectively (P < .01). The FS errors had significant clinical impact on 0.9% of the 803 patients due to insufficient resection. (https://ascopubs.org/doi/10.1200/JCO.2015.63.4907)  If it is inconclusive, the decision must be taken by the surgeon in charge who can wait for the definitive anatomy and if the resection was not sufficient, reoperate on the patient or perform an anatomical resection assuming that it may be a benign lesion. 

Lines 218-373: As this is an article for a special issue on the diagnosis and staging of NSCLC, the authors have decided to summarise everything involved in this technique in order to make it didactic, including the anatomical limits and how to position the patient. Not all readers will be surgeons with extensive experience in this type of surgery. This is not an article that pretends to be new, as this technique has been performed for years, but a little like a small summary for those who want to start performing it or who are not thoracic surgeons and want to understand these procedures. 

As to whether we recommend staging by uniportal VATS or by mediastinoscopy, the authors wanted to explain the technique and that it is feasible. The surgical indication is not global, each patient may or may not have an indication for one or the other procedure. We have explained in which type of stations it is better to perform uniportal VATS as in territory 5 or 6. The final decision will always be made by the surgeon in charge, as well as by the multidisciplinary tumour committee of each hospital and each specific case.

Round 2

Reviewer 1 Report

I have not further comments

Author Response

Thanks

Reviewer 2 Report

Thank you for your kind response and for adding the suggested revisions

Author Response

thanks to you

Reviewer 3 Report

All issues are addressed adequately, however, the explications given to the reply to the reviewers must be incorporated in the text 

Author Response

Added